# Giant piezoelectric voltage coefficient in grain-oriented modified PbTiO$_3$ material

Yongke Yan[1,2], Jie E. Zhou[3], Deepam Maurya[1], Yu U. Wang[3] & Shashank Priya[1,2]

A rapid surge in the research on piezoelectric sensors is occurring with the arrival of the Internet of Things. Single-phase oxide piezoelectric materials with giant piezoelectric voltage coefficient ($g$, induced voltage under applied stress) and high Curie temperature ($T_c$) are crucial towards providing desired performance for sensing, especially under harsh environmental conditions. Here, we report a grain-oriented (with 95% <001> texture) modified PbTiO$_3$ ceramic that has a high $T_c$ (364 °C) and an extremely large $g_{33}$ (115 × 10$^{-3}$ Vm N$^{-1}$) in comparison with other known single-phase oxide materials. Our results reveal that self-polarization due to grain orientation along the spontaneous polarization direction plays an important role in achieving large piezoelectric response in a domain motion-confined material. The phase field simulations confirm that the large piezoelectric voltage coefficient $g_{33}$ originates from maximized piezoelectric strain coefficient $d_{33}$ and minimized dielectric permittivity $\varepsilon_{33}$ in [001]-textured PbTiO$_3$ ceramics where domain wall motions are absent.

[1] Center for Energy Harvesting Materials and Systems (CEHMS), Virginia Tech, Blacksburg, Virginia 24061, USA. [2] Institute for Critical Technology and Applied Science (ICTAS), Virginia Tech, Blacksburg, Virginia 24061, USA. [3] Department of Materials Science and Engineering, Michigan Tech, Houghton, Michigan 49931, USA. Correspondence and requests for materials should be addressed to Y.Y. (email: yanthu@vt.edu) or to S.P. (email: spriya@vt.edu).

The arrival of the Internet of Things is generating opportunities for smart sensors that can operate in varying environmental conditions with ultrahigh performance[1,2]. A piezoelectric sensor utilizes the piezoelectric effect to measure changes in strain or force, acoustic pressure, and acceleration, by converting the mechanical energy into an electrical charge. Piezoelectric sensors have advantage of operating over a wide range of frequency, providing an excellent linearity over a range of input mechanical amplitude. Furthermore, they are insensitive to electromagnetic fields and radiation and can perform reliably under harsh environmental conditions (temperature, pressure and corrosion)[3]. The value of piezoelectric voltage coefficient ($g$, induced voltage under applied stress) represents material figure of merit for piezoelectric sensors[3,4].

Most of the state-of-the-art piezoelectric materials are based on perovskite-structured ferroelectrics, such as $BaTiO_3$, $PbTiO_3$ (denoted as PT), $Pb(Zr,Ti)O_3$ (denoted as PZT) and $Pb(Mg_{1/3}Nb_{2/3})O_3$-$PbTiO_3$ (denoted as PMN-PT). Among them, PZT-based piezoelectric ceramics have been widely utilized due to their superior piezoelectric performance and they can be easily tailored to meet the requirements for different application through compositional modifications[4]. The piezoelectric voltage coefficient $g_{33}$ of PZT ceramics is usually in the range of 20 to $30 \times 10^{-3}\,V\,m\,N^{-1}$, as shown in Fig. 1a. The $<001>$ oriented relaxor-$PbTiO_3$ ferroelectric single crystals have ultrahigh piezoelectric strain coefficient $d_{33}$ and electromechanical coupling factor $k_{33}$ on the order of $2,000\,pC\,N^{-1}$ and 0.9, respectively[5,6]. However, its $g_{33}$ coefficient is still $<40 \times 10^{-3}\,V\,m\,N^{-1}$. Considering the relation between $d$ and $g$ ($g = d/\varepsilon$), it is challenging to have higher value of $g$, because, any increase in piezoelectric response $d$ is usually accompanied by even larger increase in the dielectric permittivity $\varepsilon$.

To achieve high $g_{33}$, the most widely used method has been fabrication of piezoelectric composites containing high $d$ piezoelectric material (such as PZT ceramic, PMN-PT single crystal) with low $\varepsilon_r$ polymers (such as epoxy, polyvinylidene fluoride (PVDF))[7]. It is worth mentioning that PVDF polymer itself has very large $g_{33}$ due to a very small $\varepsilon_r$ ($d_{33} = 33\,pC\,N^{-1}$, $\varepsilon_r = 13$, yielding $g_{33} = 286.7 \times 10^{-3}\,V\,m\,N^{-1}$)[8]. However, the application of such piezoelectric composites and PVDF is limited to the temperature regime below the melting temperature (166 °C)[8]. Furthermore, it is difficult to integrate polymeric materials with

other functional materials or component normally synthesized by thin/thick film fabrication process requiring high temperature.

Prior studies have shown that $d_{33}$ and $g_{33}$ of piezoelectric ceramics can be simultaneously improved by a cost-effective texturing process called templated grain growth (TGG)[9–12]. For example, the piezoelectric charge/strain coefficient $d_{33}$ of $<001>$ textured PMN-PT and PMN-PZT ceramics was found to exceed $1,000\,pC\,N^{-1}$, which is about two to five times higher than that of the non-textured ceramics[9,10]. The $g_{33}$ magnitude also increased by a factor of two compared to that of the non-textured ceramics, as shown in Fig. 1a. The increase of $d_{33}$ was attributed to engineered domain state in $<001>$ textured ceramics in similar fashion as that of $<001>$ oriented single crystal. In a rhombohedral single crystal, the domain configurations consisting of the equivalent $<111>$ polarizations exhibit high piezoelectric response along the $<001>$ direction. The enhanced $g_{33}$ was related to the reduced dielectric constant of textured materials due to the presence of templates with low dielectric permittivity[9,10]. This phenomenon is analogous to the high $g_{33}$ obtained in piezoelectric single crystal–polymer composite. Although $<001>$ textured PMN-PT and textured PMN-PZT ceramics exhibit relatively large $g_{33}$ than that of their non-textured counterparts, the temperature range of application is limited by phase transition between rhombohedral and tetragonal phases ($T_{R\text{-}T}$)[13–15], as shown in Fig. 1b. The $g_{33}$ of these textured ceramics was also much lower than that of piezoelectric ceramic/single crystal–polymer composite.

In this work, we provide fundamental insight in the design of high $g_{33}$–high $T_c$ material by considering anisotropy/composition/phase structure selection, tailored microstructure and domain engineering. First, we select tetragonal PT as starting composition and phase. PT has high Curie temperature ($T_c = 490$ °C), small dielectric constant, and large piezoelectric anisotropy[16]. It has been widely used for high-temperature–high-frequency sensor and transducer applications. However, due to its large crystal anisotropy (tetragonality $c/a = 1.064$), it is very difficult to sinter PT ceramics. It is also challenging to pole pure PT ceramics due to low resistivity and high coercivity. To overcome these challenges, several dopants have been attempted and it has been shown that the mechanical properties and electrical resistance of PT ceramic can be improved via Sm and Mn doping[17]. Here we report a Sm and Mn modified PT ceramic

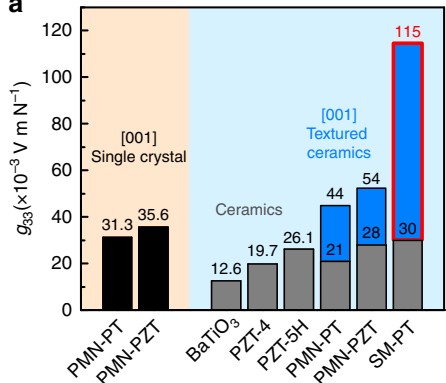
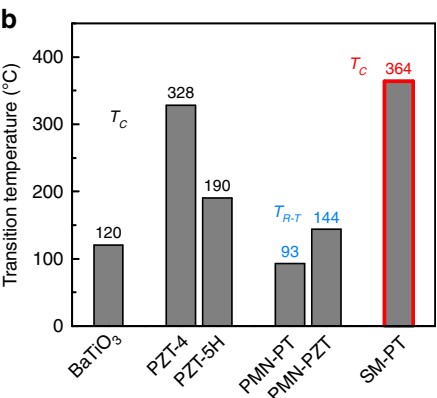

**Figure 1 | Comparison of piezoelectric voltage coefficient and phase transition temperature.** (**a**) Piezoelectric voltage coefficient ($g_{33}$) for well-known perovskite structured piezoelectric oxide single crystal (PMN-PT[13] and PMN-PZT[13]) and ceramics ($BaTiO_3$ (ref. 4), PZT-4 (ref. 4), PZT-5H[4], PMN-PT[9], PMN-PZT[10] and SM-PT). Black bars: single crystal; gray bars: ceramics; blue bars: textured ceramics; blue bar with red border: textured SM-PT in this study. (**b**) phase transition temperature above room temperature ($T_{R\text{-}T}$: rhombohedral-to-tetragonal ferroelectric phase transition; $T_c$: Curie temperature) for well-known perovskite structured piezoelectric oxide materials ($BaTiO_3$ (ref. 4), PZT-4 (ref. 4), PZT-5H[4], PMN-PT[13], PMN-PZT[13] and SM-PT). Blue bar with red border: textured SM-PT in this study.

(denoted as SM-PT) with grains textured along <001> crystallographic direction to achieve a high $T_c$ (364 °C) and an extremely large $g_{33}$ ($115 \times 10^{-3}$ Vm N$^{-1}$) in comparison with other known single-phase oxide materials, as shown in Fig. 1.

## Results

**Synthesis of textured ceramics.** The textured SM-PT ceramic was fabricated by TGG method. In this process, PT plate-like template crystals were aligned in SM-PT ceramic matrix powder by the tape casting method. During sintering, the SM-PT matrix grains grew from aligned PT templates and resulted in textured/grain-oriented SM-PT ceramics. The perovskite PT templates were synthesized using topochemical conversion method. Direct synthesis of PT templates with high aspect ratio morphology is difficult due to its cubic symmetry at the synthesis temperature (1,050 °C). To fabricate the perovskite PT template, layered perovskite structured $PbBi_4Ti_4O_{15}$ (PBiT) precursors were first synthesized and then converted into perovskite structured PT. Layered structured PBiT microcrystal can be easily grown into plate-like shape due to its strong structural anisotropy. Figure 2a shows the scanning electron microscopy (SEM) images of the PBiT precursors and PT templates after the conversion reaction. The PBiT precursors had a plate-like high-aspect ratio with a diameter of around 10 μm and a thickness of around 0.3 μm. The final product PT template was able to maintain the platelet shape inherited from the PBiT precursor due to their topotactic relationship. Figure 2b shows the X-ray diffraction patterns of

synthesized PBiT precursors and PT templates. It can be seen that PBiT can be converted into $PbTiO_3$ by the topochemical chemical conversion reaction with a little impurity phase denoted by the extra peak near 31° ($2\theta$). It should be noted here that the synthesis of pure PT is much more difficult than $BaTiO_3$, $NaNbO_3$ and so on, using same topochemical conversion method[18]. Poterala et al.[19] have systematically studied the reaction process of the PBiT in a $NaCl/Bi_2O_3/PbO$ flux system. The results show that the presence of Na$^+$ in the reaction flux will facilitate the perovskite phase formation because the incorporation of Na$^+$ charge balances Bi$^{3+}$ on the perovskite A-site (forming $PbTiO_3$-$Na_{0.5}Bi_{0.5}TiO_3$ solid solution).

Figure 2c shows the microstructure evolution and the texture development during sintering in SM-PT ceramics with 5 wt% PT seeds. It can be seen that PT templates were extremely well aligned in SM-PT matrix having particle size in the range of 200–300 nm. The high template-to-matrix grain size ratio (more than 20) is desirable for achieving large driving force for TGG. On increasing temperature up to 1,000 °C, the matrix grains start to nucleate on the templates, which eventually leads to the TGG at further higher temperatures. Although the PT template had a slight composition difference from the SM-PT, this compositional difference was homogenized via elemental diffusion at high temperature (which is the advantage of the reactive TGG), as evidenced by the uniform contrast in the back scattered electron–SEM image (Fig. 2c) and single dielectric peak in the dielectric spectra (Fig. 3a). On increasing temperature up to 1,250 °C, all the randomly oriented matrix grains disappeared

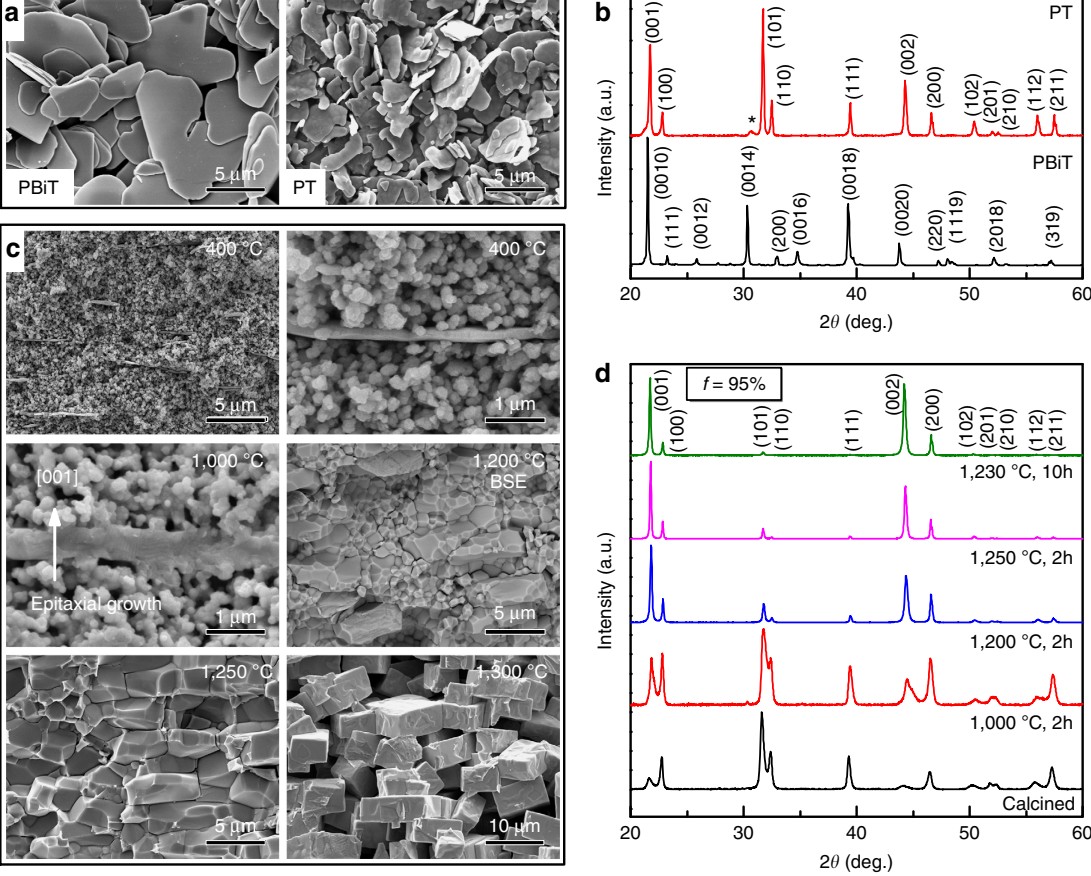

**Figure 2 | Microstructures and crystal structures of templates and textured ceramics.** (**a**) SEM images of PBiT precursors and PT templates. (**b**) X-ray diffraction patterns of synthesized PBiT precursors and PT templates. (**c**) Backscattered electron (BSE)-SEM images of PT textured SM-PT ceramics sintered at different temperature for 2 h. (**d**) X-ray diffraction patterns of calcined SM-PT matrix powders and textured SM-PT ceramics sintered at different temperature and soaking time.

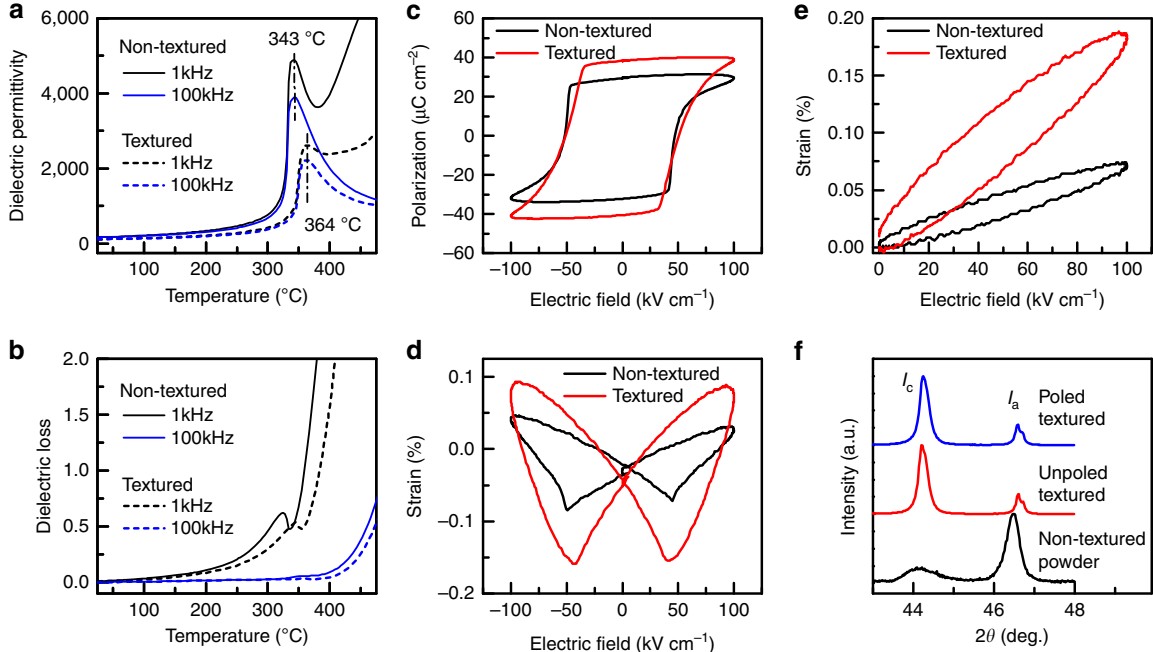

**Figure 3 | Dielectric, ferroelectric and piezoelectric properties.** (**a**) Dielectric permittivity and (**b**) dielectric loss as a function of temperature for non-textured and textured SM-PT samples. The vertical dash-dot lines are used for indicating Curie temperature. (**c**) Polarization, (**d**) bipolar strain, and (**e**) unipolar strain as a function of electric field at 75 °C for non-textured and textured samples. (**f**) X-ray diffraction patterns of non-textured matrix, unpoled and poled textured ceramics, where $I_c$ is the intensity of (002) peak and $I_a$ is the intensity of (200)/(002) peaks.

**Table 1 | Dielectric, ferroelectric and piezoelectric properties of nontextured, textured PT ceramics and PT single crystal.**

| Samples | $\varepsilon_r$ | $\tan\delta$ (1 kHz) | $T_c$ (°C) | $d_{33}$ (pC N$^{-1}$) | $d_{31}$ (pC N$^{-1}$) | $g_{33}$ ($\times 10^{-3}$ VmN$^{-1}$) | $g_{31}$ ($\times 10^{-3}$ VmN$^{-1}$) | $g_{33}/g_{31}$ | Ref. |
|---|---|---|---|---|---|---|---|---|---|
| La,Mn-doped PbTiO$_3$ ceramic | 170 | 0.008 | 470 | 51 | 4.4 | 34 | 2.9 | 11.7 | 36 |
| Sm,Mn-doped PbTiO$_3$ ceramic | 196 | 0.009 | 321 | 59 | 1.7 | 34 | 1.0 | 34 | 37 |
| PbTiO$_3$ single crystal | 125 | — | — | 143 | 26.9 | 129 | 24 | 5.4 | 38 |
| PbTiO$_3$ single crystal | 126 | — | — | 117 | 25 | 105 | 22 | 4.8 | 39 |
| Non-textured ceramic | 202 | 0.010 | 343 | 53 | 5.8 | 30 | 3.2 | 9.4 | This study |
| 82% textured ceramic | 146 | 0.013 | 364 | 95 | 8.6 | 74 | 6.7 | 11 | This study |
| 95% textured ceramic | 124 | 0.010 | 364 | 127 | 26.8 | 115 | 24.4 | 4.7 | This study |

leaving well-oriented templated grains. On further increasing the temperature, the sample was over-sintered and the density decreased due to higher porosity. The X-ray diffraction patterns in Fig. 2d confirm the texture development during the sintering process. The intensity of the $(110)_{pc}$ (pc: pseudocubic, parent phase) peak was the highest for the randomly oriented (or non-textured) ceramics. With increasing degree of texture, the intensity of $(110)_{pc}$ Bragg reflections continuously decreased while the intensity of $(001)_{pc}$ and $(002)_{pc}$ reflections increased, manifesting strong preferred crystallographic orientation along $<001>_{pc}$. To achieve high density and high texture, final samples were sintered at 1,230 °C for 10 h. The samples showed 99% relative density and 95% texture degree in terms of Lotgering factor[20].

**Enhanced piezoelectric properties with high Curie temperature.** Figure 3a shows the dielectric permittivity as a function of temperature for non-textured and textured SM-PT samples. The modification with Sm and Mn decreased the $T_c$ of PbTiO$_3$ from 490 °C to 343 °C for the non-textured sample. The textured ceramic had a little higher $T_c$ (364 °C) due to slightly less

concentration of Sm and Mn (template had no Sm and Mn). However, the Curie temperature of SM-PT ceramic was much higher than most of the PZT based ceramics (PZT-4: 328 °C; PZT-5H: 193 °C)[4]. The dielectric permittivity of textured samples was found to be lower than that of their non-textured counterparts. From Fig. 3b, it can also be seen that the dielectric loss of textured samples is slight lower than that of non-textured sample. Furthermore, it can be observed that both the samples had higher dielectric losses especially at low frequency and high temperature. Previous studies have indicated that pure PT had high dielectric losses but Mn doping was found to significantly increase the resistivity and reduce the dielectric losses. However, oxygen vacancies become mobile at high temperature, thereby, contributing to the dielectric losses. Figure 3c shows the polarization–electric field (P–E) hysteresis plots for the non-textured and textured samples. It can be seen that the polarization of the textured samples was higher than that of the non-textured samples. Theoretically, the intrinsic polarization value $P_s$ along the polar axis of the mono-domain crystal for the tetragonal phase follows the relationship $P_{s,<001>} = (3)^{1/2} P_{s,<111>}$ (ref. 21). Due to the averaging of polarization in three-dimensional space, the non-textured

ceramic has a $P$ between that of $<001>$ and $<111>$ textured ceramics. It can be seen that the polarization values $P_s$ derived from the measured hysteresis loops are well consistent with the theoretical estimation.

Table 1 summarizes the dielectric, ferroelectric and piezo-electric properties of the non-textured and textured samples. It can be seen that the piezoelectric strain coefficient $d_{33}$ was increased from 53 pC N$^{-1}$ (non-textured ceramics) to 127 pC N$^{-1}$ in textured ceramics. More importantly, a large magnitude of $g_{33}$ (115 × 10$^{-3}$ Vm N$^{-1}$) was obtained in textured samples, which was significantly higher than that of PT, PZT and PMN-PT ceramics or single crystals as shown in Fig. 1a.

**Anisotropy of dielectric and piezoelectric properties.** Figure 4a shows the dependence of dielectric and piezoelectric properties on the degree of texture. With the increase in $<001>$ texture degree, it can be found that the $d_{33}$ increases, $\varepsilon_r$ decreases, resulting in a large magnitude of $g_{33}$ in textured samples.

To better understand the effects of crystallographic orientation of grains on the piezoelectric properties, the orientation dependence was calculated using structural relationships. Using spherical coordinates for 4mm tetragonal crystal, the longitudinal dielectric permittivity and piezoelectric strain coefficient as a function of angle $\theta$ away from the polar axis is given as (refs 22,23):

$$\varepsilon_r^* = \varepsilon_{r,11} \sin^2 \theta + \varepsilon_{r,33} \cos^2 \theta, \quad (1)$$

$$d_{33}^* = \cos \theta (d_{31} \sin^2 \theta + d_{15} \sin^2 \theta + d_{33} \cos^2 \theta). \quad (2)$$

The values for PbTiO$_3$ were taken from ref. 24. Figure 5 shows the orientation dependence of dielectric permittivity and piezoelectric

strain coefficient. It can be seen that $\varepsilon_r$ has the minimum value along [001] direction while $d_{33}$ has the maximum value along [001] direction. On the basis of the relation $g = d/\varepsilon_r$, the $g_{33}$ is maximized along [001] direction.

Interestingly, the PbTiO$_3$ shows the maximum value of $d_{33}$ along its polar axis, while the widely studied morphotropic phase boundary composition PMN-PT and PZT and even tetragonal BaTiO$_3$ show their largest piezoelectric magnitude along non-polar direction. PbTiO$_3$ is tetragonal below the Curie temperature without any intermediate ferroelectric–ferroelectric phase transitions. Because of the absence of a proximal phase transition, the shear coefficient $d_{15}/d_{33}$ of PbTiO$_3$ is small, and the contribution of polarization rotation is very weak. A large $d_{15}/d_{33}$ is related to proximity to ferroelectric–ferroelectric phase transitions due to flattening of the free energy function whether induced by changes in composition or temperature, or by application of an electric field or stress[25]. These results suggest that the mechanism for enhanced piezoelectric response in PbTiO$_3$ is polarization extension, which is different from polarization rotation observed in PMN-PT, PZT and BaTiO$_3$ systems.

It should be noted that for PT materials, two types of anisotropy need to be considered. The first type of anisotropy refers to the magnitude change of a particular parameter, such as $d_{33}$, $g_{33}$, $\varepsilon_r$ as a function of crystallographic orientation. As mentioned above, with increase in $<001>$ texture degree, the value of $d_{33}$ increases, $\varepsilon_r$ decreases, and consequently $g_{33}$ increases. The second type of piezoelectric anisotropy is the ratio of $g_{33}/g_{31}$. As listed in Table 1, the distinct macroscopic piezoelectric anisotropy (high $g_{33}/g_{31}$ or $d_{33}/d_{31}$) of the doped PT random ceramics may be not a property of the single crystal but of the ceramic[26,27]. In modified PT ceramics, the ratio of $d_{33}/d_{31}$ (or $g_{33}/g_{31}$) has been reported to be over 10; however, this ratio ($d_{33}/d_{31}$) is about 4.7 for a PT single crystal. Figure 4b shows the ratio of $g_{33}/g_{31}$ in $<001>$ textured PT ceramics as a function of texture degree, where it can be found that the ratio of $g_{33}/g_{31}$ is about 10 at low texture degree, but rapidly decreases to 4.8 when the ceramic is highly textured (more than 0.9). In this study, the new finding is the record-breaking value of large $g_{33}$, and our focus is to maximize $g_{33}$ based on crystallographic anisotropy (first type) through maximized ratio of $d_{33}$ and $\varepsilon_r$, not $g_{33}/g_{31}$. Nevertheless, it is worth noting that a combination of both large $g_{33}$ (about 70 × 10$^{-3}$ Vm N$^{-1}$) and large $g_{33}/g_{31}$ (about 11) can be achieved at partial texture around $f = 0.8$, which is an attractive performance.

**Self-polarization and domain alignment.** In addition to the calculation of intrinsic piezoelectric anisotropy (Fig. 5) of tetra-gonal PbTiO$_3$ single crystal based on polarization rotation, we further investigated the effect of domain switching and domain wall motion on piezoelectric response in SM-PT ceramics. Domain switching and domain wall motion is generally used to explain the extrinsic contribution towards the piezoelectric response in polycrystalline ceramics. Figure 3d,e display the bipolar and unipolar electric field induced strains of non-textured and textured sample at 75 °C. The unipolar strains at 100 kV cm$^{-1}$ are only 0.19% and 0.07%, for textured and non-textured ceramics, respectively, which suggests the absence of 90° domain switching. If 90° domain were switchable, SM-PT could have exhibited large electric field-induced strain around 6% in textured ceramics and around 2.5% in non-textured ceramics as shown by phase field model below. Li et al.[28] used a combined theoretical and experimental approach to establish a relation between crystallographic symmetry and the ability of a ferroelectric polycrystalline ceramic to switch, and found that

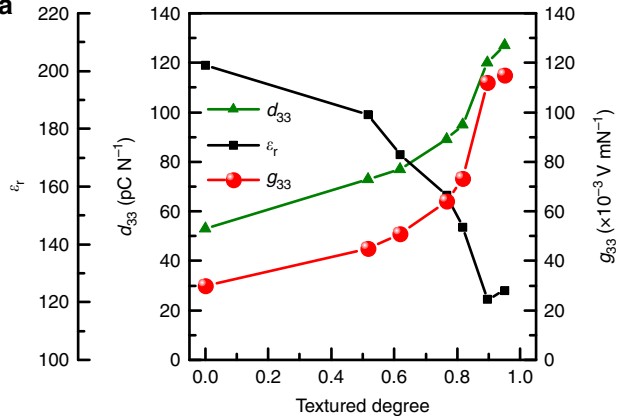

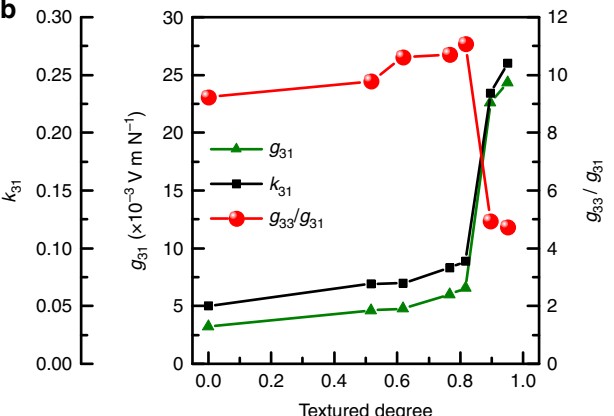

**Figure 4 | Piezoelectric properties of doped PT ceramics as a function of texture degree.** (a) $\varepsilon_r$, $d_{33}$ and $g_{33}$ and (b) $k_{31}$, $g_{31}$ and $g_{33}/g_{31}$.

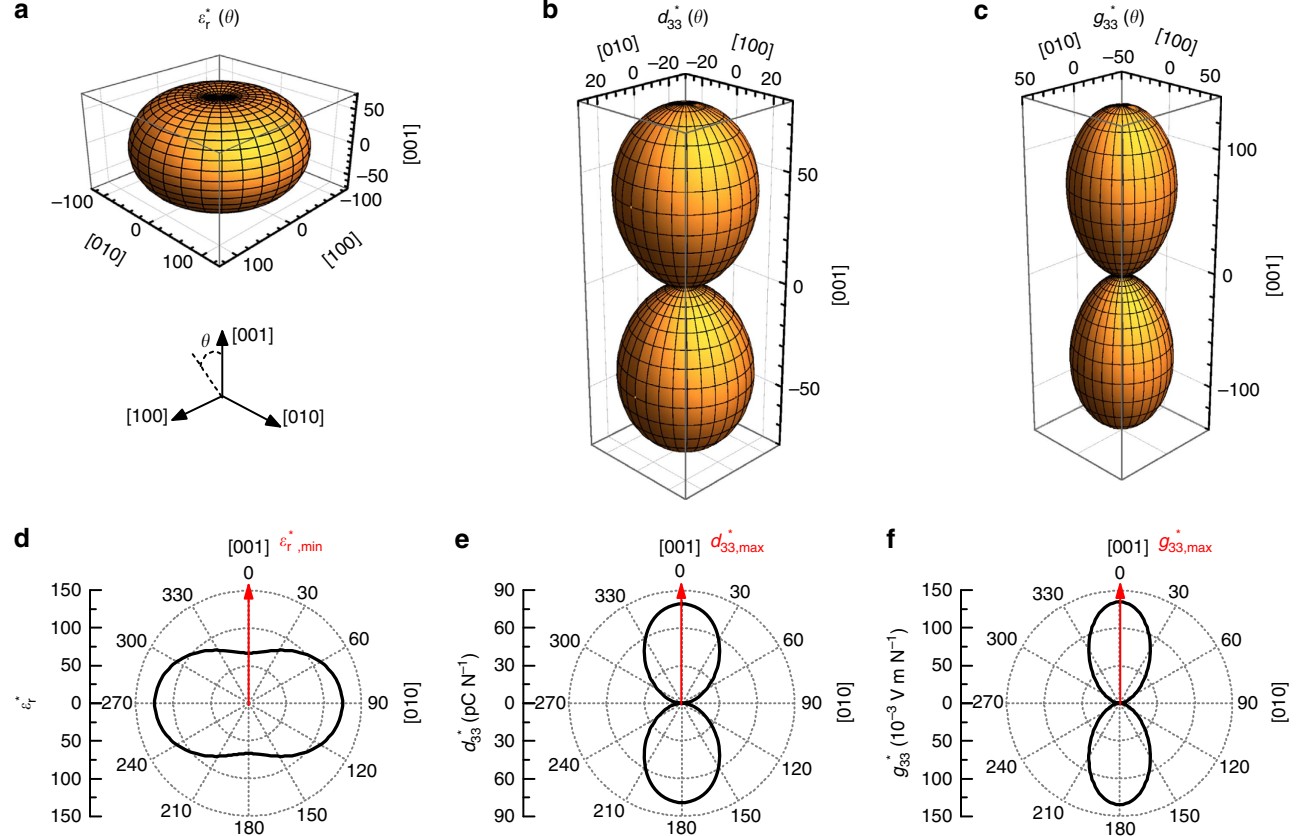

**Figure 5 | Anisotropy of dielectric and piezoelectric properties.** Orientation dependence of (**a**) dielectric permittivity $\varepsilon_r^\star(\theta)$, (**b**) piezoelectric strain coefficient $d_{33}^\star(\theta)$, and (**c**) piezoelectric voltage coefficient $g_{33}^\star(\theta)$ of tetragonal PbTiO$_3$ crystal. (**d**) Cross-section of the surface of (**a**); (**e**) cross-section of the surface of (**b**); (**f**) cross-section of the surface of (**c**). The coordinate system indicates crystallographic axes. Numbers on axes indicate values of $\varepsilon_r^\star(\theta)$, $d_{33}^\star(\theta)$, $g_{33}^\star(\theta)$, respectively.

an equiaxed tetragonal polycrystal will not show 90° domain switching and macroscopic strains through domain switching.

To understand the non-180° domain switching, the X-ray diffraction patterns were recorded on poled and unpoled samples. Due to large tetragonality of PT ceramics, $(002)_{pc}$ (pc: pseudocubic, parent phase) peak splits into two peaks, (002) and (200). The relative intensity ratio of (002) and (200) peaks indicates the percentage of $c$-domain and $a$-domain. On the basis of the X-ray diffraction pattern shown in Fig. 3f, it can be seen that the percentage of $c$-domain and $a$-domain was not changed under electric field during poling process. Furthermore, for non-textured sample, the theoretical intensity ratio of (002) and (200) is 1:2. Interestingly, we noticed that the percentage of $c$-domain ($I_c$) in textured ceramic is much larger than $a$-domain ($I_a$). This phenomenon indicates that the <001>-textured SM-PT ceramic exhibits a strong polarization self-alignment or $c$-domain preferred orientation.

**Observation of domain switching**. To experimentally observe the domain motion under electric field, vertical and lateral piezoresponse force microscopy (PFM) was performed for both non-textured and textured samples, as shown in Fig. 6. Several inferences can be drawn from the observations in this figure: (a) From the amplitude in vertical mode, it can be seen that the amplitude of non-textured sample has much higher contrast than textured sample due to the wider orientation distribution of each grains; (b) From the phase in vertical mode, it can be seen that 180° domain switching occurred in both non-textured and

textured sample; (c) From amplitude and phase in lateral mode, it can be seen that <001> textured sample has much weaker piezoresponse than non-textured samples; (d) Combined the phase contrast from vertical and lateral modes, it can be found that there is no 90° domain motion and only 180° domain switching occurred in both non-textured and textured samples. The difficulty in 90° domain switching can be attributed to the significantly higher activation energy for the non-180° domain switching in tetragonal PbTiO$_3$, as discussed earlier. High activation energy of 90° domain switching can be also indicated from the domain structures of textured SM-PT ceramic under transmission electron microscopy (TEM) as shown in Supplementary Fig. 1 and described in Supplementary Note 1.

**Phase field model of non-textured and textured SM-PT**. To further quantitatively investigate the domain-level mechanisms for the enhanced piezoelectric voltage coefficient in [001]-textured PbTiO$_3$ ceramics, we adopted a phase field model for ferroelectrics (see Supplementary Information for details)[29–33]. To analyse the mechanisms for enhanced piezoelectric voltage coefficient in [001]-textured PbTiO$_3$ ceramics, we simulated both non-textured and [001]-textured Pb(Zr$_{1-x}$Ti$_x$)O$_3$ polycrystals with the composition of $x = 1$ (PT) and $x = 0.6$ (PZT) at room temperature for comparison. PZT with $x = 0.6$ has an equilibrium tetragonal phase as PT does but is closer to the morphotropic phase boundary and has a reduced electrocrystalline anisotropy than PT, thus serving as a good case study for comparison. The same grain structure

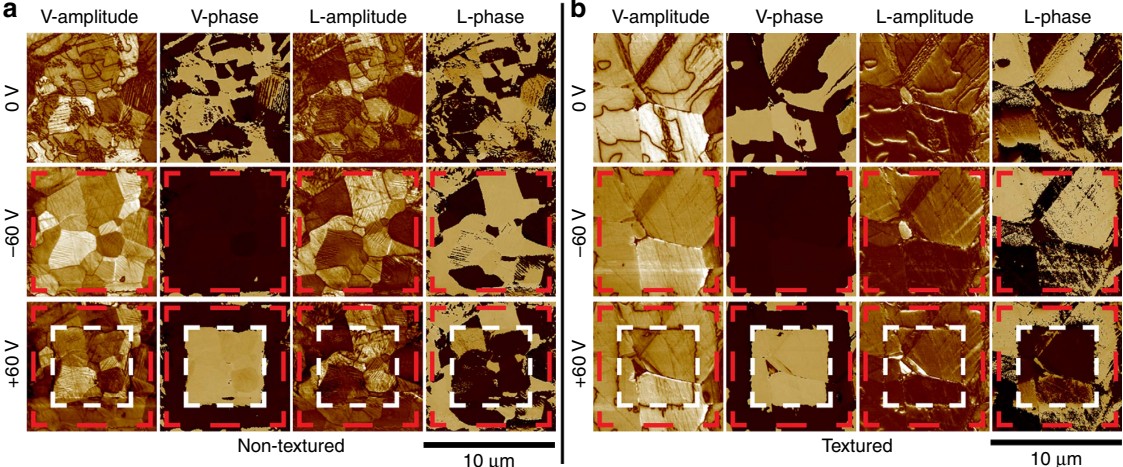

**Figure 6 | Domain structures and domain switching.** PFM images of (**a**) nontextured and (**b**) textured ceramics. The vertical (V) and lateral (L) PFM was measured under off resonance, $f = 17$ kHz. The dimension of each scanning area is $10 \times 10 \ \mu$m. The square area with red dash line border was poled at DC bias of $-60$ V; the square area with white dash line border was poled at $+60$ V.

(Supplementary Fig. 2a) and the two different textures (Supplementary Fig. 2b,c) are used in the simulations to exclude other varying factors that could complicate the comparison study. The simulation system is discretized into $512 \times 512$ computational grids with periodic boundary conditions. The piezoelectric voltage coefficient $g_{ij}$ can be evaluated through either of the direct and converse piezoelectric effects using $g_{ij} = -(\partial E_i/\partial \sigma_j)_{\mathbf{D}}$ and $g_{ij} = (\partial \varepsilon_j/\partial D_i)_{\boldsymbol{\sigma}}$, respectively, which are equivalent thermodynamic definition via Maxwell's relation[34]. It is relatively easier to implement the stress free condition ($\boldsymbol{\sigma} = 0$) in the phase field simulation, thus we simulated the converse piezoelectric effect to evaluate $g_{33} = \partial \varepsilon_3/\partial D_3 = d_{33}/\varepsilon_{33}$ through the piezoelectric strain coefficient $d_{33}$ and dielectric permittivity $\varepsilon_{33}$.

The simulated polarization distributions and domain structures in non-textured and textured ceramics of PZT and PT that are poled in vertical direction were compared (Supplementary Fig. 3). In the non-textured ceramics shown in Supplementary Fig. 3a,b, both PZT and PT have polarization distributions significantly deviated from the poling direction, as expected for ceramics with random grain orientations. Nevertheless, PT possesses dominantly tetragonal phase, while PZT possesses significant fraction of rhombohedral phase and smaller fraction of orthorhombic phase that coexists with the tetragonal phase, as shown in Supplementary Fig. 4a,b. The non-equilibrium rhombohedral and orthorhombic phase distortions are caused by internal electric field and stress. Such internal fields are present in both non-textured PZT and PT, but the phase distortion is prominent only in PZT due to its significantly reduced electrocrystalline anisotropy with its composition closer to the morphotropic phase boundary. In contrast, in the textured ceramics shown in Supplementary Fig. 3c,d, both PZT and PT have polarization distributions aligned in the poling direction, as expected for [001] texturing where grain orientation is in the [001] poling direction. Slight deviation of polarization vectors from the poling direction is observed as shown in dimmer colors, which is caused by imperfect [001]-texturing. In the simulations the [001] axes of the grains are distributed within a cone of 5° half-apex angle. PT has larger polarization than PZT, as shown by longer vectors. Both textured PZT and PT possess only tetragonal phase as shown in Supplementary Fig. 4c,d, in contrast to phase coexistence in non-textured ceramics shown in Supplementary Fig. 4a,b. It is worth noting that while phase coexistence in non-textured PZT ceramics helps accommodate the electrostriction strain associated

with non-uniform polarization distribution, PT has very stable tetragonal phase with lattice strain as large as about 6% that cannot be accommodated in non-textured ceramics thus often causing cracks in real samples (cracking is not considered in the computer simulations, where the very large internal stress instead causes local non-equilibrium phase distortion, which is different from real non-textured PT ceramics).

Figure 7 compares the simulated polarization-electric field and strain-electric field curves in non-textured and textured PZT and PT ceramics. It is observed that, while domain switching is easy in non-textured PZT, the coercive field in textured PZT is significantly increased, which makes domain switching more difficult. On the other hand, domain switching in PT for both non-textured and textured ceramic is always difficult due to the large coercive field. It is worthwhile to mention that in the computer simulation a very large electric field (above $1,600 \, \text{kV cm}^{-1}$) was applied on textured PT to observe domain switching without considering dielectric breakdown or mechanical cracking, which is not possible in real samples. Domain switching is also difficult in non-textured PT, since the applied electric field is usually below its coercive field (above $400 \, \text{kV cm}^{-1}$). As discussed earlier, if switchable, PT could exhibit large electric field-induced strain around 6% in textured ceramics and around 2.5% in non-textured ceramics, which were not observed as shown in Fig. 3d,e.

Figure 8 compares the simulated piezoelectric strain coefficient, dielectric permittivity, and piezoelectric voltage coefficient in non-textured and textured PZT and PT ceramics. The value of $d_{33}$ and $\varepsilon_{33}$ were obtained from the simulated strain-electric field and polarization-electric field responses, respectively, from which the piezoelectric voltage coefficient was evaluated using relation, $g_{33} = d_{33}/\varepsilon_{33}$. The simulated value of $130 \times 10^{-3} \, \text{Vm N}^{-1}$ agrees well with the experimental value of $115 \times 10^{-3} \, \text{Vm N}^{-1}$, thus confirming the giant piezoelectric voltage coefficient in [001]-textured PT ceramics. To further reveal the underlying mechanisms of such a significant enhancement in $g_{33}$, simulations were also performed with domain walls frozen by setting the gradient coefficient $\beta = 0$ in Supplementary Equation 1, which produces sharp domain walls that are pinned by discrete computational grids to imitate pinning effects. It is found that domain wall motions contribute about 50% to both $d_{33}$ and $\varepsilon_{33}$ in non-textured PT ceramics, thus have negligible effect on $g_{33}$. On the other hand, domain wall motions do not play a role in

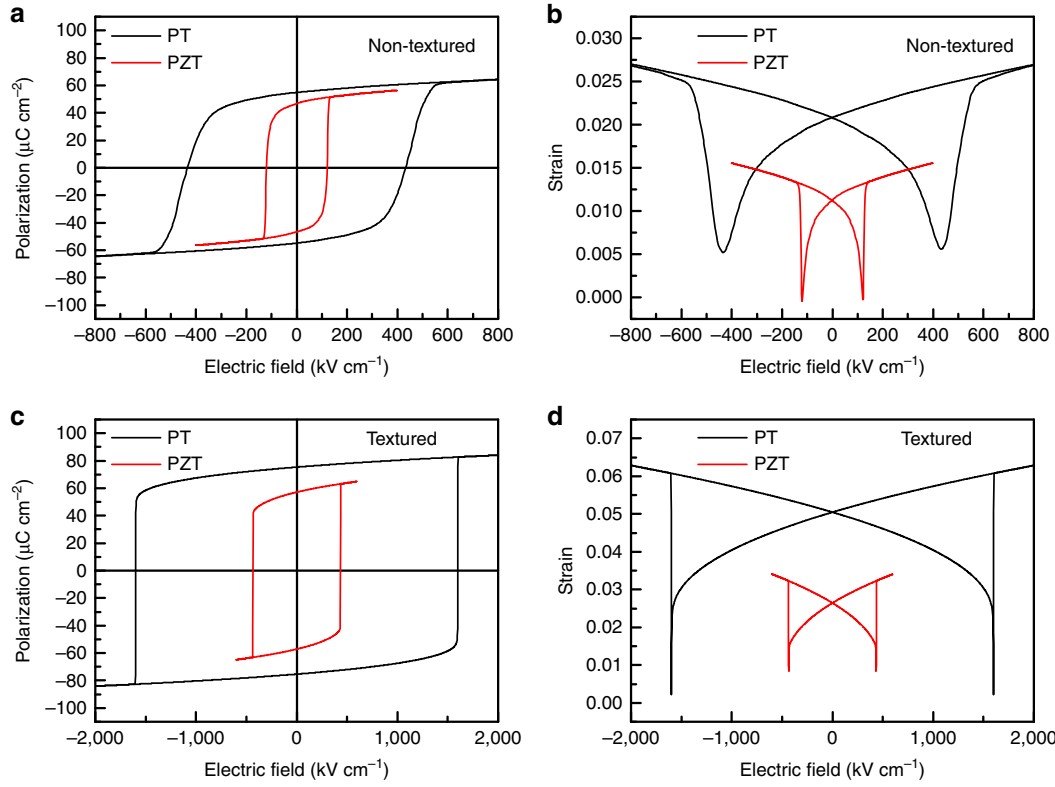

**Figure 7 | Phase field simulated polarization-electric field and strain-electric field curves.** (**a**) polarization-electric field curves of non-textured PZT and PT ceramics; (**b**) strain–electric field curves of non-textured PZT and PT ceramics; (**c**) polarization-electric field curves of textured PZT and PT ceramics; (**d**) strain–electric field curves of textured PZT and PT ceramics.

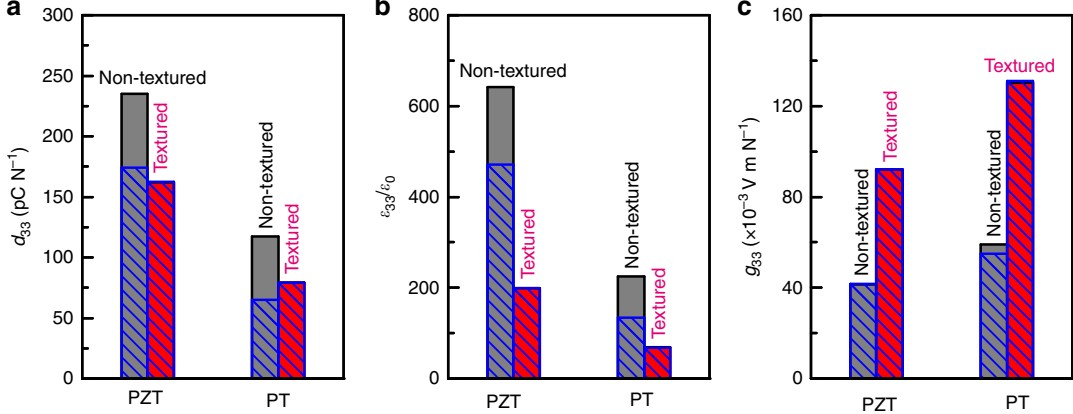

**Figure 8 | Phase-field-simulated piezoelectric and dielectric properties.** (**a**) piezoelectric strain coefficient ($d_{33}$), (**b**) dielectric permittivity ($\varepsilon_{33}/\varepsilon_0$), and (**c**) piezoelectric voltage coefficient ($g_{33}$) in non-textured (gray bars) and textured (red bars) PZT and PT ceramics. To reveal contributions from domain wall motions, values of these coefficients are also plotted with domain walls frozen in the simulations, as shown by hatched blue bars.

textured PT ceramics, because polarizations are well aligned in [001] direction. The comparison study between textured and non-textured PT ceramics shows that PT has an intrinsically high value of $d_{33}$ and low value of $\varepsilon_{33}$ along [001] axis due to its high electrocrystalline anisotropy, which is in agreement with the experimental measurements shown in Fig. 4 and theoretical analysis shown in Fig. 5. Therefore, the [001]-texturing effectively reduces $\varepsilon_{33}$ resulting in a significant enhancement in $g_{33}$ of [001]-textured PT ceramics. The simulations show that similar mechanisms also improve $g_{33}$ in textured PZT, but the achievable value is less than in textured PT. Thus, appropriate choice of PT composition combined with texture engineering and domain

engineering effectively controls the grain structures and domain processes in [001]-textured PT ceramics to achieve giant piezo-electric voltage coefficient, which provides rational design and synthesis of piezoelectric materials for targeted applications.

## Discussion

We report a grain-oriented (with 95% <001> texture) modified PbTiO₃ material with high $T_c$ (364 °C) and an extremely large $g_{33}$ (115 × 10⁻³ Vm N⁻¹) as compared with other known single-phase oxide materials. Diffraction and scanning probe micro-scopy studies reveal that self-polarization due to grain orientation

along the spontaneous polarization direction plays an important role in achieving large piezoelectric response in a domain motion-confined material. The simulations using phase field model confirm that the large piezoelectric voltage coefficient $g_{33}$ originates from maximized piezoelectric strain coefficient $d_{33}$ and minimized dielectric permittivity $\varepsilon_{33}$ in [001]-textured $PbTiO_3$ ceramics where domain wall motions are absent.

## Methods

**Sample preparation.** The composition of SM-PT was $(Pb_{0.8725}Sm_{0.085})$ $(Ti_{0.98}Mn_{0.02})O_3$. The SM-PT matrix powder was synthesized by conventional solid state reaction method. For this, PbO (99.9%, Sigma-Aldrich), $Sm_2O_3$ (99.9%, Alfa Aesar), $TiO_2$ (Ishihara Sangyo Kaisha Ltd.), and $MnO_2$ (99.9%, Alfa Aesar) was mixed and ball-milled in ethanol for 24 h. The mixture was dried at 80 °C and then calcined at 850 °C for 4 h. The calcined powders were ball-milled again with 1.5 wt% excess PbO for 24 h. The templates for texturing SM-PT ceramic are plate-like $<001>$ $PbTiO_3$ microcrystals. The $<001>$ $PbTiO_3$ templates were synthesized by topochemical microcrystal conversion method. In the first step, $PbBi_4Ti_4O_{15}$ precursor was synthesized in molten salt. In the next step, $PbBi_4Ti_4O_{15}$ precursor was mixed with PbO and NaCl salt, and then heated to 1,050 °C for 3 h. $Bi^{3+}$ in $PbBi_4Ti_4O_{15}$ was substituted by the $Pb^{2+}$ from PbO, yielding $PbTiO_3$ template and $Bi_2O_3$ byproduct. The $Bi_2O_3$ byproduct was removed by diluted nitric acid. To fabricate textured ceramics, the ceramic slurry was prepared by ball milling the SM-PT matrix powders with organic binder (Ferro 73225), and toluene/ethanol solvents. Next 5 wt% of PT templates were mixed into the slurry by magnetic stirring. Afterwards, the slurry was casted at the rate of 40 cm min$^{-1}$ by using doctor blade with height of 250 μm. The dried green tapes were cut, stacked, and laminated at 80 °C under 20 MPa pressure for 15 min. The green samples were heated to 400 °C for 2 h with a heating rate of 0.3 °C min$^{-1}$ to remove organic solvent and binder, and then isostatically pressed at 200 MPa for 1 min. Samples were subsequently sintered at 1,000–1,300 °C for 2–10 h. The detailed process for the synthesis of grain oriented/textured ceramics is provided elsewhere[35].

**Characterization.** The phase and microstructure were characterized using X-ray diffraction (D8 Advanced, Bruker) and SEM (FEI Quanta 600 FEG, Philips). The degree of pseudo-cubic $<001>$ texture was determined from the X-ray diffraction pattern in $2\theta$ range of 20–60° by Lotgering factor method. The dielectric properties of poled samples were measured as a function of temperature by using a multi-frequency LCR meter (HP4274A). The piezoelectric properties of samples were obtained by resonance and anti-resonance technique using impedance/gain phase analyzer (HP 4194A) and $d_{33}$-meter (YE 2730 A, APC Products, Inc., PA). Piezoresponse force microscopy (PFM, Bruker Dimension Icon) was used to image the ferroelectric domain structures. Conductive Platinum-Iridium silicon cantilevers (SCM-PIT, Bruker) were used for the PFM characterization. Standard grinding and ion-milling method was used to prepare the electron transparent TEM specimens, and FEI Titan 300 microscope was used to capture TEM images.

**Data availability.** The data that support the findings of this study are available from the corresponding authors on request.

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

## Acknowledgements

We gratefully acknowledge the financial supports from AFOSR (S.P., FA9550-14-1-0376), AFOSR STTR program (Y.Y.), National Science Foundation (D.M., PHY-1242637), and DOE (J.Z. and Y.W., DE-FG02-09ER46674). The parallel computer simulations were performed on XSEDE supercomputers. The authors would like to thank

the Center for Energy Harvesting Materials and Systems for providing access to industrial experts and equipment.

## Author contributions

Y.Y. and S.P. conceived the idea; Y.Y. prepared samples, conducted X-ray diffraction, SEM, PFM characterization and performed dielectric/ferroelectric/piezoelectric measurement; J.E.Z. & Y.U.W. conducted modelling analysis; D.M. performed TEM characterization and analysis; S.P. supervised the research. All authors contributed discussion and revised the manuscript; Y.Y., Y.U.W. and S.P. wrote the manuscript.

## Additional information

**Competing financial interests:** The authors declare no competing financial interests.

