## [Peer Review File · Nature Communications]

Reviewers' comments:

Reviewer #1 (Remarks to the Author):

Piezoelectric voltage coefficient (g) represents the parameter in considering the materials for sensors. The g_{33} of PZT ceramics is usually in the range of $20\sim 30 \times 10^{-3}$ Vm/N, and that of the $\langle 001 \rangle$ oriented relaxor-PbTiO₃ ferroelectric single crystals is still less than 40×10^{-3} Vm/N. PVDF polymer has very large g_{33} of 286.7×10^{-3} Vm/N due to a very small ϵ_r ($d_{33} = 33$ pC/N, $\epsilon_r = 13$), however the application of such piezoelectric composites and PVDF is limited to the temperature regime below the melting temperature (166 {degree sign}C). The authors prepared a grain-oriented (with 95% $\langle 001 \rangle$ texture) modified-PbTiO₃ material that has a high T_c (~ 364 oC) and an extremely large g_{33} (115×10^{-3} Vm/N). This work is very important to develop high-quality piezoelectric sensors.

My questions include:

- 1) For the sensor applications, the piezoelectric voltage coefficient is of great importance. With regards to the authors' statement in Page 3. I am wondering that why the most popular piezoelectric component is PZT or PMN-PT oxides, rather than the PVDF?
- 2) The most interesting point in this manuscript is the large value of piezoelectric voltage coefficient, but the authors try to interpret the issue based on traditional "4mm tetragonal crystal model". Is that possible to understand the underlying physics using a precise modeling?
- 3) The question in Page 11 "why does the PT show the maximum?" is an interesting one. However, to my best knowledge, I really think the claim "because of the absence of proximal phase transition, the shear coefficient" is not correct. Since, the T-phase PMN-xPT ($x > 38$) and PZN-xPT ($x > 11$) had no intermediate FE-FE phase transition below T_c , but the maximum value of d_{33} was obtained along its nonpolar direction $\langle 111 \rangle$.
- 4) Some Figures and discussion about domains should be kept in supporting information since they are not the most important for the conclusion of this paper.

Reviewer #2 (Remarks to the Author):

The manuscript by Y. Yan et al. is of interest and gives new impacts in the field of modern piezoelectric materials. The summary of the key results of this manuscript consists in the characterisation of the novel grain-oriented (with 95% $\langle 001 \rangle$ texture) modified-PbTiO₃ material that exhibits the high Curie temperature and high piezoelectric sensitivity (the g_{33} value, longitudinal piezoelectric effect). In this context, the manuscript will be of value and interest in the field of piezoelectric materials and their sensor applications.

The degree of originality and scientific merit are high. It should be mentioned that many problems appear at manufacturing the pure PbTiO₃ ferroelectric ceramic, however its modification enables one to improve the microstructure, properties and stability of the ceramic medium. The high degree of texture is of merit additionally.

The experimental data and supplement show the high reserach level of authors and their good abilities to represent the new results in the clear form. The quality of data is high,

however some pertinent additions are to be made (see below). The quality of presentation in general corresponds to the journals of the 'Nature' group. The experimental data are given in the appropriate form and should be taken into account in future piezotechnical (sensor) applications. In the manuscript, no word "conclusion" has been found.

The following comments are to be taken by the authors into consideration.

(i) The conclusion (10-15 lines, not longer) is to be added in pp.20-21.

(ii) As is known, modified PbTiO₃-type ferroelectric ceramics are characterised by the large piezoelectric anisotropy [see, for instance, the review paper by Turik A.V. and Topolov V.Yu., J Phys D Appl Phys 30:1541-1549 (1997)]. However, the piezoelectric anisotropy is mainly considered for PbTiO₃ single-domain crystal (pp.10-11) without a due comparison to the poled ferroelectric ceramics. It should be clearly shown that the large ratio of $g_{33} / |g_{31}|$ is typical of the novel textured material put forward by Yan et al. If experimental data are available, then it would be good to add a dependence of the anisotropy $g_{33} / |g_{31}|$ (or the same for d_{3j}) on the degree of texture.

(iii) The new experimental results are to be compared to the data on modified PbTiO₃ ceramic; see, e.g., Ikegami S. et al., J Acoust Soc Am 50:1060-1066 (1971). This paper contains data on d_{3j} and ϵ_{33} , and therefore, one can evaluate g_{3j} for comparison to those in the manuscript.

(iv) In Table 1.1, p.9 of monograph "Piezo-active composites. Orientation effects..." by Topolov et al. (Springer, 2014), there are experimental data on the single-domain PbTiO₃ single crystal, e.g., $d_{33} = 143$ pC / N and ϵ_{33} -free = 125. Based on them, one can evaluate g_{33} . This value would be comparable to the experimental result $g_{33} = 115$ mV m / N on the highly-textured material from the reviewed paper. Such a comparison would be useful after the authors' analysis of the anisotropic state in terms of the single-domain single crystal (pp.10-11 of the manuscript).

In general, the level of clarity in the manuscript is good, but the due improvement is needed.

In conclusion, I recommend to publish the manuscript in "Nature Communications" after the suggested changes, in the improved form.

Reviewers' comments:

Reviewer #1 (Remarks to the Author):

Piezoelectric voltage coefficient (g) represents the parameter in considering the materials for sensors. The g_{33} of PZT ceramics is usually in the range of $20\sim 30\times 10^{-3}$ Vm/N, and that of the $\langle 001 \rangle$ oriented relaxor-PbTiO₃ ferroelectric single crystals is still less than 40×10^{-3} Vm/N. PVDF polymer has very large g_{33} of 286.7×10^{-3} Vm/N due to a very small ϵ_r ($d_{33}= 33$ pC/N, $\epsilon_r = 13$), however the application of such piezoelectric composites and PVDF is limited to the temperature regime below the melting temperature (166 °C). The authors prepared a grain-oriented (with 95% $\langle 001 \rangle$ texture) modified-PbTiO₃ material that has a high T_c (~ 364 °C) and an extremely large g_{33} (115×10^{-3} Vm/N). This work is very important to develop high-quality piezoelectric sensors. My questions include:

- (1) For the sensor applications, the piezoelectric voltage coefficient is of great importance. With regards to the authors' statement in Page 3. I am wondering that why the most popular piezoelectric component is PZT or PMN-PT oxides, rather than the PVDF?

Answer: We agree with the reviewer that the most popular piezoelectric material is PZT or PMN-PT oxides, rather than the PVDF, based on the consideration of (a) useable temperature range and environmental stability, (b) processing challenges that require complex poling process such as corona poling, and (c) flexibility in implementation. In addition to the main limitation of operating temperature range and temperature stability which is determined by the Curie temperature. It is generally accepted that commercial PVDF devices should not be used above 65 – 80 °C because of deterioration of the piezoelectric performance. In poling PVDF, high electrical field and mechanical stretching (stabilization of β phase) are needed simultaneously to align the dipoles to achieve polarization. Figure R1-R3 shows the issues with temperature stability of piezoelectric, ferroelectric, mechanical properties of PVDF and copolymer PVDF-TrFE.¹

Response Figure R1 | Change in the d_{33} coefficient with the annealing temperature.¹

Response Figure R2 | Change in P_r with the temperature at a driving field of 130 MV/m.¹

Response Figure R2 | Effect of the temperature on the d_{31} coefficients and storage moduli (E') of the PVDF homopolymer bimorph and PVDF-TrFE copolymer bimorph: (a) unannealed, (b) annealed for 24 h at 110 °C, and (c) annealed for 24 h at 140 °C (T_g : glass-transition temperature).¹

Response Figure R4 | Temperature ranges of electronics for different applications.²

Current industrial trends are focused on electronics that can operate reliably in harsh environments, including extremely high temperatures. Figure R4 shows the market trends and drivers for high temperature electronic components for different industries.² From the considerations of operating temperature range and long term stability, PZT and PT piezoelectric ceramics show significant advantages over PVDF polymer.

(2) The most interesting point in this manuscript is the large value of piezoelectric voltage coefficient, but the authors try to interpret the issue based on traditional “4mm tetragonal crystal model”. Is that possible to understand the underlying physics using a precise modeling?

Answer: More precise modeling does exist, which provides a microscopic understanding of the underlying physics, and the findings agree with the experimental observations and the “traditional” Landau-Ginzburg-Devonshire theory where the material parameters are derived from the experiments. First-Principles Density Functional Theory has been employed to carry out computational study of ferroelectric perovskite compounds including PT.³⁻⁸ These studies show that the underlying mechanisms responsible for the piezoelectric and dielectric properties of PT are attributed to the electronic structure, covalent-ionic character of atomic bonding, and hybridization of orbitals between different atoms. In particular, PT behaves as “extender” ferroelectric, in contrast to “rotator” ferroelectric (e.g., BaTiO₃).⁹ Such behaviors are well

described by the “traditional” thermodynamic treatment based on Landau-Ginzburg-Devonshire theory since the coefficients and material parameters are fitted to reproduce the experimental results. Therefore, our seemingly traditional interpretation does have a solid foundation supported by more precise modeling such as First-Principles Density Functional Theory.

- (3) The question in Page 11 “why does the PT show the maximum?” is an interesting one. However, to my best knowledge, I really think the claim “because of the absence of proximal phase transition, the shear coefficient” is not correct. Since, the T-phase PMN-xPT ($x>38$) and PZN-xPT ($x>11$) had no intermediate FE-FE phase transition below T_c , but the maximum value of d_{33} was obtained along its nonpolar direction $\langle 111 \rangle$.

Answer: While the fundamental reason must be explained from microscopic mechanisms at electronic level, as mentioned in our above answer to Comment 2, proximal inter-ferroelectric phase transformation also provides useful insights into this interesting question¹⁰: perovskite-type ferroelectric compounds (e.g., BaTiO₃) and pseudo-binary solid solutions (e.g., PMN-PT and PZN-PT) do exhibit similar thermodynamic behaviors in the sense of FE-FE phase transitions, wherein the former is induced by temperature while the latter is induced by composition (across morphotropic phase boundary); there are two or more FE phases in the vicinity of inter-ferroelectric phase transformation temperature or composition, which changes the ferroelectric anisotropy by flattening the free energy landscape, promoting the shear coefficient. In this sense, PT is different: as the end member of the pseudo-binary solid solutions, PT is far from MPB and there is no intermediate FE-FE phase transition below T_c , therefore PT does not show flattening of free energy function. On the other hand, the properties of PMN-PT and PZN-PT are attributed to the existence of MPB and its effects on free energy function and ferroelectric anisotropy.

- (4) Some Figures and discussion about domains should be kept in supporting information since they are not the most important for the conclusion of this paper.

Answer: We agree with the reviewer. we have moved the previous Figure 5 [Figure 5 | Domain structures of textured SM-PT ceramic under TEM] and discussion about domains into the supporting information. We only kept previous Figure 6 [Figure 6 | PFM images of nontextured

and textured ceramics], because this data indicated the difficulty of 90° domain switching in tetragonal PbTiO₃, which bridges the experimental macroscopic piezoelectric properties and modeling.

Reviewer #2 (Remarks to the Author):

The manuscript by Y. Yan et al. is of interest and gives new impacts in the field of modern piezoelectric materials. The summary of the key results of this manuscript consists in the characterization of the novel grain-oriented (with 95% <001> texture) modified-PbTiO₃ material that exhibits the high Curie temperature and high piezoelectric sensitivity (the g_{33} value, longitudinal piezoelectric effect). In this context, the manuscript will be of value and interest in the field of piezoelectric materials and their sensor applications.

The degree of originality and scientific merit are high. It should be mentioned that many problems appear at manufacturing the pure PbTiO₃ ferroelectric ceramic, however its modification enables one to improve the microstructure, properties and stability of the ceramic medium. The high degree of texture is of merit additionally.

The experimental data and supplement show the high research level of authors and their good abilities to represent the new results in the clear form. The quality of data is high; however some pertinent additions are to be made (see below). The quality of presentation in general corresponds to the journals of the 'Nature' group. The experimental data are given in the appropriate form and should be taken into account in future piezotechnical (sensor) applications. In the manuscript, no word "conclusion" has been found.

In general, the level of clarity in the manuscript is good, but the due improvement is needed.

In conclusion, I recommend to publish the manuscript in "Nature Communications" after the suggested changes, in the improved form.

The following comments are to be taken by the authors into consideration.

- (1) The conclusion (10-15 lines, not longer) is to be added in pp.20-21.

Answer: We have added Conclusion in the revised manuscript following the reviewer's comment. On page of 13, we have added the following text in red color into the main manuscript:

“In summary, we report a grain-oriented (with 95% $\langle 001 \rangle$ texture) modified-PbTiO₃ material with high T_c (~ 364 °C) and an extremely large g_{33} (115×10^{-3} Vm/N) as compared to the other known single phase oxide materials. Diffraction and scanning probe microscopy studies reveal that self-polarization due to grain orientation along the spontaneous polarization direction plays an important role in achieving large piezoelectric response in a domain-motion-confined material. The simulations using phase field model confirm that the large piezoelectric voltage coefficient g_{33} originates from maximized piezoelectric strain coefficient d_{33} and minimized dielectric permittivity ϵ_{33} in [001]-textured PbTiO₃ ceramics where domain wall motions are absent.”

(2) As is known, modified PbTiO₃-type ferroelectric ceramics are characterised by the large piezoelectric anisotropy [see, for instance, the review paper by Turik A.V. and Topolov V.Yu., J Phys D Appl Phys 30:1541-1549 (1997)]. However, the piezoelectric anisotropy is mainly considered for PbTiO₃ single-domain crystal (pp.10-11) without a due comparison to the poled ferroelectric ceramics. It should be clearly shown that the large ratio of $g_{33} / |g_{31}|$ is typical of the novel textured material put forward by Yan et al. If experimental data are available, then it would be good to add a dependence of the anisotropy $g_{33} / |g_{31}|$ (or the same for d_{3j}) on the degree of texture.

Answer: In revised manuscript, we added the dependence of the anisotropy g_{33}/g_{31} on the degree of texture. We have also provided some additional piezoelectric properties, such as ϵ_r , d_{33} , g_{33} , k_{31} , g_{31} , as a function of the texture degree. These data suggests that we need to consider two types of anisotropy.

(a) Crystallographic anisotropy

This type of anisotropy refers to the magnitude change of a particular parameter, such as d_{33} , g_{33} , ϵ_r as a function of crystallographic orientation. As shown in Response Figure R5a, with the increase in $\langle 001 \rangle$ texture degree, the d_{33} increases, ϵ_r decreases, and g_{33} increases. We utilized crystallographic anisotropy to achieve the goal of large g_{33} , as we discussed in the main manuscript.

Response Figure R5 | Piezoelectric properties of doped PT ceramics as a function of texture degree: (a) ϵ_r , d_{33} and g_{33} and (b) k_{31} , g_{31} and g_{33}/g_{31} .

(b) Macroscopic piezoelectric property anisotropy, g_{33}/g_{31}

Figure R5b shows the ratio of g_{33}/g_{31} in $\langle 001 \rangle$ textured PT ceramics as a function of texture degree. It can be found that the ratio of g_{33}/g_{31} is about 10 at low texture degree, but rapidly decreases to 4.8 when the ceramic is highly textured (e.g. >0.9). Based on the literature, the distinct macroscopic piezoelectric anisotropy (high g_{33}/g_{31} or d_{33}/d_{31}) of the doped PT random ceramics may not be a property of the single crystal but of the ceramic.¹¹ In modified (such as Ca or Sm) PT ceramic, the ratio of d_{33}/d_{31} has been reported to be over 10.¹² In PT single crystal, d_{33}/d_{31} is only 4.7, as measured by Gavrilachenko.¹¹ As suggested by experimental results and phase field modeling, $\langle 001 \rangle$ textured PT has multi-domain structure but 90° domain switching is difficult, thus the ratio of g_{33}/g_{31} in highly $\langle 001 \rangle$ -textured PT ceramics is similar to single-domain crystals.

In our study, the giant value of g_{33} was realized by exploiting crystallographic anisotropy (first type) through maximized ratio of d_{33} and ϵ_r . Further, a combination of both large $g_{33} \sim 70 \times 10^{-3}$ Vm/N and large $g_{33}/g_{31} \sim 11$ can be achieved at partial texture around $f = 0.8$, which is also attractive for applications.

In the manuscript, we made the changes as marked by red color:

Anisotropy of dielectric and piezoelectric properties. Figure 4(a) shows the dependence of dielectric and piezoelectric properties on the degree of texture. With increase in $\langle 001 \rangle$ texture degree, it can be found that d_{33} increases while ϵ_r decreases, resulting in a large magnitude of g_{33} in textured samples.

To better understand the effects of crystallographic orientation of grains on the piezoelectric properties, the orientation dependence was calculated using structural relationships. Using spherical coordinates for 4mm tetragonal crystal, the longitudinal dielectric permittivity and piezoelectric strain coefficient as a function of angle θ away from the polar axis is given as^{22,23}:

$$\epsilon_r^* = \epsilon_{r,11} \sin^2 \theta + \epsilon_{r,33} \cos^2 \theta, \quad (1)$$

$$d_{33}^* = \cos \theta (d_{31} \sin^2 \theta + d_{15} \sin^2 \theta + d_{33} \cos^2 \theta), \quad (2)$$

The values for PbTiO_3 were taken from Ref. 24. Figure 5 shows the orientation dependence of dielectric permittivity and piezoelectric strain coefficient. It can be seen that ϵ_r has the minimum value along $[001]$ direction while d_{33} has the maximum value along $[001]$ direction. Based on the relation $g = d/\epsilon_r$, the g_{33} is maximized along $[001]$ direction.

The question is “Why does the PbTiO_3 show the maximum value of d_{33} for PbTiO_3 along its polar axis while the widely studied MPB composition in PMN-PT and PZT and even tetragonal BaTiO_3 show their largest piezoelectric magnitude along non-polar direction?” PbTiO_3 is tetragonal below the Curie temperature without any intermediate ferroelectric-ferroelectric phase transitions. Because of the absence of a proximal phase transition, the shear coefficient d_{15}/d_{33} of PbTiO_3 is small, and the contribution of polarization rotation is very weak. A large d_{15}/d_{33} is related to proximity to ferroelectric-ferroelectric phase transitions due to flattening of the free energy function whether induced by changes in composition or temperature, or by

application of an electric field or stress²⁵. These results suggest that the tetragonal PbTiO₃ has different mechanism for enhanced piezoelectric response, ‘polarization extension dominant’, which can be distinguished from that of PMN-PT, PZT and BaTiO₃, where the mechanism is ‘polarization rotation dominant’.

It should be noted that for PT materials, two types of anisotropy needed to be considered. The first type of anisotropy refers to the magnitude change of a particular parameter, such as d_{33} , g_{33} , ϵ_r as a function of crystallographic orientation. As mentioned above, with increase in $\langle 001 \rangle$ texture degree, the value of d_{33} increases, ϵ_r decreases, and consequently g_{33} increases. The second type of piezoelectric anisotropy is the ratio of g_{33}/g_{31} . As listed in Table 1, the distinct macroscopic piezoelectric anisotropy (high g_{33}/g_{31} or d_{33}/d_{31}) of the doped PT random ceramics may be not a property of the single crystal but of the ceramic^{30,31}. In modified PT ceramics, the ratio of d_{33}/d_{31} (or g_{33}/g_{31}) has been reported to be over 10, however, this ratio (d_{33}/d_{31}) is about 4.7 for a PT single crystal. Figure 4(b) shows the ratio of g_{33}/g_{31} in $\langle 001 \rangle$ textured PT ceramics as a function of texture degree, it can be found that the ratio of g_{33}/g_{31} is about 10 at low texture degree, but rapidly decreases to 4.8 when the ceramic is highly textured (>0.9). In this study, the new finding is the record-breaking value of large g_{33} , and our focus is to maximize g_{33} based on crystallographic anisotropy (first type) through maximized ratio of d_{33} and ϵ_r , not g_{33}/g_{31} . Nevertheless, it should be noted that a combination of both large $g_{33} \sim 70 \times 10^{-3}$ Vm/N and large $g_{33}/g_{31} \sim 11$ can be achieved at partial texture around $f = 0.8$, which is attractive performance.

- (c) The new experimental results are to be compared to the data on modified PbTiO₃ ceramic; see, e.g., Ikegami S. et al., J Acoust Soc Am 50:1060-1066 (1971). This paper contains data on d_{3j} and ϵ_{33} , and therefore, one can evaluate g_{3j} for comparison to those in the manuscript.

Answer: We have summarized these data in the table 1 for comparison.

Table 1 | Dielectric, ferroelectric and piezoelectric properties of nontextured, textured PT ceramics and PT single crystal.

Samples	ϵ_r	$\tan\delta$ (1 kHz)	T_c (°C)	d_{33} (pC/N)	d_{31}	g_{33} ($\times 10^{-3}$ Vm/N)	g_{31}	g_{33}/g_{31}	Ref.
La,Mn doped PT ceramic	170	0.008	470	51	4.4	34	2.9	11.7	[22]

Sm,Mn doped PT ceramic	196	0.009	321	59	1.7	34	1.0	34	[23]
PbTiO ₃ single crystal	125	-	-	143	26.9	129	24	5.4	[24]
PbTiO ₃ single crystal	126	-	-	117	25	105	22	4.8	[25]
non-textured ceramic	202	0.010	343	53	5.8	30	3.2	9.4	this work
82% textured ceramic	146	0.013	364	95	8.6	74	6.7	11	this work
95% textured ceramic	124	0.010	364	127	28.3	115	24.4	4.7	this work

(d) In Table 1.1, p.9 of monograph "Piezo-active composites. Orientation effects..." by Topolov et al. (Springer, 2014), there are experimental data on the single-domain PbTiO₃ single crystal, e.g., $d_{33} = 143$ pC / N and ϵ_{33} -free = 125. Based on them, one can evaluate g_{33} . This value would be comparable to the experimental result $g_{33} = 115$ mV m / N on the highly-textured material from the reviewed paper. Such a comparison would be useful after the authors' analysis of the anisotropic state in terms of the single-domain single crystal (pp.10-11 of the manuscript).

Answer: We have summarized these data in the table 1 for comparison.

Table 1 | Dielectric, ferroelectric and piezoelectric properties of nontextured, textured PT ceramics and PT single crystal.

Samples	ϵ_r	$\tan\delta$ (1 kHz)	T_c (°C)	d_{33} (pC/N)	d_{31}	g_{33} ($\times 10^{-3}$ Vm/N)	g_{31}	g_{33}/g_{31}	Ref.
La,Mn doped PT ceramic	170	0.008	470	51	4.4	34	2.9	11.7	[22]
Sm,Mn doped PT ceramic	196	0.009	321	59	1.7	34	1.0	34	[23]
PbTiO ₃ single crystal	125	-	-	143	26.9	129	24	5.4	[24]
PbTiO ₃ single crystal	126	-	-	117	25	105	22	4.8	[25]
non-textured ceramic	202	0.010	343	53	5.8	30	3.2	9.4	this work
82% textured ceramic	146	0.013	364	95	8.6	74	6.7	11	this work
95% textured ceramic	124	0.010	364	127	28.3	115	24.4	4.7	this work

Reference

1. Dargaville T. R, et al., Evaluation of piezoelectric poly(vinylidene fluoride) polymers for use in space environments. I. temperature limitations, *Journal of Polymer Science Part B: Polymer Physics*, **43**, 1310-1320 (2005).

2. Bultitude J. et al., Development and Characterization of Novel Leaded High Temperature Multi-Layer Ceramic Capacitors (MLCC) made using Transient Liquid Phase Sintering (TLPS) Materials, MS&T 15, Columbus, OH, Oct 5, 2015.
3. Cohen R.E. Origin of ferroelectricity in perovskite oxides, *Nature*, **358**, 136 (1992).
4. King-Smith R.D., Vanderbilt D., First-principles investigation of ferroelectricity in perovskite compounds, *Physical Review B*, **49**, 5828 (1994).
5. Saghi-Szabo G., Cohen R.E., Krakauer H., First-principles study of piezoelectricity in PbTiO_3 , *Physical Review Letters*, **80**, 4321 (1998).
6. Saghi-Szabo G., Cohen R.E., Krakauer H., First-principles study of piezoelectricity in tetragonal PbTiO_3 and $\text{PbZr}_{1/2}\text{Ti}_{1/2}\text{O}_3$, *Physical Review B*, **59**, 12771 (1999).
7. Fu H. and Cohen R.E., Polarization rotation mechanism for ultrahigh electromechanical response in single-crystal piezoelectrics, *Nature*, **403**, 281 (2000).
8. Fu H., Bellaiche L., First-principles determination of electromechanical responses of solids under finite electric fields, *Physical Review Letters*, **91**, 057601 (2003).
9. Davis M., Budimir M., Damjanovic D., Setter N., Rotator and extender ferroelectrics: Importance of the shear coefficient to the piezoelectric properties of domain-engineered crystals and ceramics, *Journal of Applied Physics*, **101**, 054112 (2007).
10. Wang Y.U., Field-induced inter-ferroelectric phase transformations and domain mechanisms in high-strain piezoelectric materials: Insights from phase field modeling and simulation, *J. Mater. Sci.*, **44**, 5225-5234, 2009.
11. Wersing W, Lubitz K, Mohaupt J. Anisotropic piezoelectric effect in modified PbTiO_3 ceramics, *IEEE T Ultrason Ferr* **36**, 424-433 (1989).
12. Turik A.V. and Topolov V.Yu., Ferroelectric ceramics with a large piezoelectric anisotropy, *J. Phys. D: Appl. Phys.* **30**:1541-1549 (1997).

REVIEWERS' COMMENTS:

Reviewer #1 (Remarks to the Author):

Piezoelectric voltage coefficient (g) represents the parameter in considering the materials for sensors. The g_{33} of PZT ceramics is usually in the range of $20\sim 30 \times 10^{-3}$ Vm/N. The authors prepared a grain-oriented (with 95% $\langle 001 \rangle$ texture) modified-PbTiO₃ material that has a high T_c (~ 364 °C) and an extremely large g_{33} (115×10^{-3} Vm/N). This work is very important to develop high-quality piezoelectric sensors.

The authors already revised the manuscript according to my previous comments, and I recommend this manuscript to be published in Nature Communication.

Reviewer #2 (Remarks to the Author):

In the revised version of the manuscript, the authors have improved the presentation of main research results and answered questions concerned with the 1st review. The key results are summarised in the good manner and can be understood by a reader. The manuscript reports new and original results that are related to the piezoelectric sensitivity of the novel PbTiO₃-based ferro- and piezoelectric material. The data are represented in the clear form, and their validity is confirmed by experimental results and evaluations. It's the good paper where the experiment coexists with due evaluations and the clear interpretation. Conclusions are formulated also good, and their reliability is high.

No improvements are needed before the publishing process. In the list of references: please check all units. For instance, in Ref. 30, no "&" is written before the last author, and in Ref. 31, we see "and" instead of "&" before the last author.

In general, the authors have taken the former review into account by 100% and really improved the paper. The clarity of the text with many new important results, appropriateness of the abstract, introduction and conclusions, good illustrations make this manuscript of interest to specialists in the field of modern piezoelectric and related materials. I recommend the present manuscript for the publication in 'Nature Communications'.

REVIEWERS' COMMENTS:

Reviewer #1 (Remarks to the Author):

Piezoelectric voltage coefficient (g) represents the parameter in considering the materials for sensors. The g_{33} of PZT ceramics is usually in the range of $20\sim 30\times 10^{-3}$ Vm/N. The authors prepared a grain-oriented (with 95% $\langle 001 \rangle$ texture) modified-PbTiO₃ material that has a high T_c (~ 364 °C) and an extremely large g_{33} (115×10^{-3} Vm/N). This work is very important to develop high-quality piezoelectric sensors.

The authors already revised the manuscript according to my previous comments, and I recommend this manuscript to be published in Nature Communication.

Response: We thank the reviewer for providing us encouraging feedback.

Reviewer #2 (Remarks to the Author):

In the revised version of the manuscript, the authors have improved the presentation of main research results and answered questions concerned with the 1st review. The key results are summarized in the good manner and can be understood by a reader. The manuscript reports new and original results that are related to the piezoelectric sensitivity of the novel PbTiO₃-based ferro- and piezoelectric material. The data are represented in the clear form, and their validity is confirmed by experimental results and evaluations. It's the good paper where the experiment coexists with due evaluations and the clear interpretation. Conclusions are formulated also good, and their reliability is high.

No improvements are needed before the publishing process. In the list of references: please check all units. For instance, in Ref. 30, no "&" is written before the last author, and in Ref. 31, we see "and" instead of "&" before the last author.

Response: We have checked and made the corrections in revised manuscript.

In general, the authors have taken the former review into account by 100% and really improved the paper. The clarity of the text with many new important results, appropriateness of the abstract, introduction and conclusions, good illustrations make this manuscript of interest to specialists in

the field of modern piezoelectric and related materials. I recommend the present manuscript for the publication in 'Nature Communications'.

Response: We thank the reviewer for providing us encouraging feedback.